# VI-116, A Novel Potent Inhibitor of VRAC with Minimal Effect on ANO1

**DOI:** 10.3390/ijms23095168

**Published:** 2022-05-05

**Authors:** Dongkyu Jeon, Kunhi Ryu, Sungwoo Jo, Ikyon Kim, Wan Namkung

**Affiliations:** College of Pharmacy, Yonsei Institute of Pharmaceutical Sciences, Yonsei University, 85 Songdogwahak-ro, Yeonsu-gu, Incheon 21983, Korea; armisael1990@gmail.com (D.J.); rkh1497@gmail.com (K.R.); dsdyu2005@naver.com (S.J.); ikyonkim@yonsei.ac.kr (I.K.)

**Keywords:** VI-116, VRAC, inhibitor, ANO1, ANO2

## Abstract

Volume-regulated anion channel (VRAC) is ubiquitously expressed and plays a pivotal role in vertebrate cell volume regulation. A heterologous complex of leucine-rich repeat containing 8A (LRRC8A) and LRRC8B-E constitutes the VRAC, which is involved in various processes such as cell proliferation, migration, differentiation, intercellular communication, and apoptosis. However, the lack of a potent and selective inhibitor of VRAC limits VRAC-related physiological and pathophysiological studies, and most previous VRAC inhibitors strongly blocked the calcium-activated chloride channel, anoctamin 1 (ANO1). In the present study, we performed a cell-based screening for the identification of potent and selective VRAC inhibitors. Screening of 55,000 drug-like small-molecules and subsequent chemical modification revealed 3,3′-((2-hydroxy-3-methoxyphenyl)methylene)bis(4-hydroxy-2H-chromen-2-one) (VI-116), a novel potent inhibitor of VRAC. VI-116 fully inhibited VRAC-mediated I^−^ quenching with an IC_50_ of 1.27 ± 0.18 μM in LN215 cells and potently blocked endogenous VRAC activity in PC3, HT29 and HeLa cells in a dose-dependent manner. Notably, VI-116 had no effect on intracellular calcium signaling up to 10 μM, which completely inhibited VRAC, and showed high selectivity for VRAC compared to ANO1 and ANO2. However, DCPIB, a VRAC inhibitor, significantly affected ATP-induced increases in intracellular calcium levels and Eact-induced ANO1 activation. In addition, VI-116 showed minimal effect on hERG K^+^ channel activity up to 10 μM. These results indicate that VI-116 is a potent and selective VRAC inhibitor and a useful research tool for pharmacological dissection of VRAC.

## 1. Introduction

Cell volume regulation is an important homeostatic function in living cells, and volume-regulated anion channel (VRAC) is the major Cl^−^ channel in regulatory volume decrease (RVD) [1]. Recently, two research groups revealed that LRRC8A is an essential protein for VRAC through whole-genome siRNA library screening [2,3]. LRRC8A is a major component of swelling-induced chloride current (*I*_Cl, swell_) that occurs during RVD in various cells. Previous studies showed that *I*_Cl, swell_ and RVD were significantly blocked when LRRC8A was downregulated by siRNA [2]. VRAC is involved not only in volume regulation but also various physiological roles, such as apoptotic volume decrease (AVD), release of excitatory amino acids (EAAs), and taurine transport [4,5]. Interestingly, VRAC is formed by heterohexamers of the LRRC8 protein family (LRRC8A−E) and is ubiquitously expressed in almost all types of mammalian cells [2,3]. Subunit composition of VRAC influences physiological processes such as taurine transport and superoxide production [5,6].

Anoctamin 1 (ANO1), also known as TMEM16A, is highly expressed in epithelial cells and other cell types such as smooth muscle cells and sensory neurons, and is involved in a variety of physiological events, such as fluid secretion, cell proliferation, migration, smooth muscle contraction and nociception [7]. VRAC plays an important role in the regulation of various pathophysiological functions, such as Pt-based anti-cancer drug resistance and glucose homeostasis through the regulation of insulin secretion [5,8,9]. Interestingly, ANO1 is also involved in insulin secretion in pancreatic beta cells, similar to VRAC, and is highly amplified in various types of tumors. ANO1 gene amplification is correlated with poor survival rate [10,11,12]. VRAC and ANO1 are co-expressed in some cell types and are mainly activated by cell swelling and intracellular calcium signaling, respectively, but they are also involved in some of the same physiological events. In particular, during swelling, VRAC and ANO1 can be activated simultaneously [7,8,9]. Therefore, in order to elucidate the physiological functions of VRAC and ANO1, selective inhibitors for each ion channel are required.

As shown in our previous study, Ani9, a potent and selective ANO1 inhibitor, showed minimal inhibition of VRAC activity at high concentrations [13]. However, in the case of VRAC inhibitors; 4,4′-Diisothiocyano-2,2′-stilbenedisulfonic acid (DIDS); 5-nitro-2, 3-(phenylpropylamino)-benzoic acid (NPPB); 4-(2-butyl-6,7-dichlor-2-cyclopentylindan-1-on-5-yl) oxobutyric acid (DCPIB); tamoxifen; mibefradil; mefloquine; clomiphene; nafoxidine and carbenoxolone; pranlukast; and zafirlukast have been identified as VRAC inhibitors, but most of these VRAC inhibitors have a low potency or low selectivity [14,15,16,17,18,19,20,21]. Although DCPIB is most often used as a potent VRAC inhibitor, DCPIB inhibits glutamate transporter GLT-1and Cx43 hemichannel and activates Twik-related K^+^ channel 1 (TREK1), TREK2, and BK channels [22,23,24]. In addition, DCPIB acutely increased intracellular calcium concentration in Panc-1 and IGR39 cells [24], and our experiments showed that DCPIB also potently inhibited ATP-induced intracellular calcium level in FRT cells.

In this study, we identified a novel potent inhibitor of VRAC, VI-116, using a cell based high-throughput screening (HTS), and investigated the effects of VI-116 and DCPIB on endogenous VRAC activity and other ion channels including ANO1, ANO2 and hERG.

## 2. Results

### 2.1. Identification of Novel Small Molecule Inhibitors of VRAC

We performed a cell-based HTS to identify potent and selective VRAC inhibitors. To measure VRAC activity, human glioma cells, LN215, were stably transfected with halide sensors YFP-F46L/H148Q/I152L. As shown in Figure 1A, to investigate the effect of compounds on VRAC activity, cells were treated with hypotonic solution (150 mOsm) containing 25 μM of test compounds for 5 min, followed by iodide (70 mM) containing solution. In Figure 1B, we investigated the effect of DCPIB, a VRAC inhibitor, on VRAC activity in LN215 cells expressing YFP-F46L/H148Q/I152L. The YFP fluorescence was significantly decreased by DCPIB with an IC_50_ of 3.45 μM in a dose-dependent manner. We screened 55,000 drug-like small-molecules and identified three novel VRAC inhibitors, VI-101, VI-201 and VI-301 (Figure 1C,D).

### 2.2. Identification of a Potent and Selective VRAC Inhibitor, VI-116

Since there are no VRAC inhibitors that potently inhibit VRAC without effects on ANO1, we further investigated the effects of three novel VRAC inhibitors on VRAC and ANO1 activity. Unfortunately, all three novel VRAC inhibitors also potently blocked both VRAC and ANO1 (Figure 2). To discover more potent and selective VRAC inhibitors, we performed a chemical modification study based on VI-101, which has the highest efficacy. We observed the effects of 19 analogs of VI-101 on VRAC and ANO1 activity and finally identified VI-116, a potent and selective VRAC inhibitor that inhibits VRAC activity (IC_50_ = 1.3 μM) with 13-fold higher potency than ANO1 activity (IC_50_ = 39.1 μM) (Table 1). Notably, the most potent analog of VI-101, VI-110, has an IC_50_ value of 0.58 μM. However, since VI-110 also potently inhibited the activity of ANO1 (IC_50_ = 4.8 μM), VI-116, a relatively potent and most selective VRAC inhibitor, was selected for further study.

### 2.3. VI-116 Potently Blocks VRAC-Mediated Chloride Currents

We further investigated the effect of VI-116 on VRAC activity using YFP quenching assay in LN215 cells. As shown in Figure 3, VI-116 (IC_50_ = 1.28 μM) more potently blocked VRAC activity, compared to DCPIB (IC_50_ = 3.45 μM). To investigate the effect of VI-116 and DCPIB on VRAC activity in the presence of 10% FBS, LN215 cells were treated with VI-116 and DCPIB in a medium containing 10% FBS for 5 min. In the presence of 10% FBS, the IC_50_ of VI-116 and DCPIB were 6.89 μM and 14.1 μM, respectively (Figure 3D). To determine whether VI-116 and DCPIB are cytotoxic, we observed the effect of VI-116 and DCPIB on cell viability in NIH-3T3 cells. As shown in Figure 3E,F, VI-116 did not affect cell viability up to 30 μM, but DCPIB significantly reduced cell viability at 30 μM.

To investigate the effect of VI-116 on *I*_Cl, swell_, we used the whole-cell voltage-clamp technique in HEK293T and LN215 cells. VI-116 completely blocked hypotonic perfusion activated *I*_Cl, swell_ in a dose-dependent manner in both HEK293T and LN215 cells (Figure 4).

### 2.4. Minimal Effect of VI-116 on Human ANO1, ANO2 and hERG Channel Activity

To investigate whether VI-116 affects ANO1 and ANO2 chloride channel activity, we first measured the effect of VI-116 on intracellular calcium signaling in FRT cells. As shown in Figure 5A, VI-116 did not affect the ATP-induced increase in intracellular calcium levels up to 10 μM and showed a weak inhibitory effect on intracellular calcium signaling at 30 μM. To observe the effect of VI-116 on ANO1 channel activity, apical membrane current of ANO1 was measured in FRT cells expressing human ANO1. Notably, VI-116 did not alter the ANO1 activation by ATP or E_act_, a specific activator of ANO1 [25], and ANO1 chloride channel was completely blocked by Ani9 (Figure 5B,C). ANO2 activity was measured using YFP fluorescence quenching assay in FRT cells expressing human ANO2. VI-116 had a minimal effect on ANO2 up to 10 μM in FRT cells expressing human ANO2 (Figure 5D).

Unexpectedly, DCPIB strongly blocked ATP-induced intracellular calcium levels increasing in a dose-dependent manner (Figure 5E). Interestingly, DCPIB had a unique effect on E_act_-activated ANO1 chloride channel activity. At 10 μM, ANO1 chloride current activated by E_act_ was increased, but at 100 μM, ANO1 chloride current was almost completely inhibited (Figure 5F).

To observe the effect of VI-116 and DCPIB on CFTR, a cAMP-regulated chloride channel, we measured the apical membrane current of CFTR in FRT cells expressing human CFTR. As shown in Figure 6A−C, VI-116 and DCPIB blocked CFTR chloride channel with IC_50_ values of 12.4 μM and 71.7 μM, respectively. We also observed the effect of VI-116 and DCPIB on human ether-a-go-go related gene (hERG) K^+^ channel, which is a major anti-target of drug discovery and induces long QT syndrome when the channel is blocked [26]. VI-116 showed minimal inhibitory effect on hERG K^+^ channel activity at 10 μM with IC_50_ of 89.6 μM, whereas DCPIB significantly blocked hERG K^+^ channel activity at 10 μM with an IC_50_ of 11.4 μM (Figure 6D–F). These results suggest that VI-116 is a potent and selective VRAC inhibitor with useful applications for in vitro and in vivo experiments.

### 2.5. VI-116 Potently Inhibits Endogenous VRAC Activity in PC3, HT29 and HeLa Cells

A heterologous complex of LRRC8A and LRRC8B−E constitutes VRAC, and VRAC channel properties differ depending on the type of heterogeneous complex [5]. Therefore, the characteristics of VRAC also depend on the cell type. Here, we investigated the effect of VI-116 on VRAC activity in three different cell lines, PC3 prostate, HT29 colon, and HeLa cervical cancer cells. As shown in Figure 7, the quantitative real-time PCR (qRT-PCR) analysis showed that PC3 cells had relatively high expression rates of LRRC8A (D and E) but HT26 cells and HeLa cells had relatively high expression rates of LRRC8A and LRRC8D, respectively. VI-116 potently and completely blocked VRAC activity in PC3, HT29, and HeLa cells with an IC_50_ of 0.63 ± 0.05 μM, 3.14 ± 1.96 μM, and 1.20 ± 0.08 μM, respectively. In the case of DCPIB, it showed weaker potency than VI-116 in PC3, HT29, and HeLa cells with an IC_50_ of 4.15 ± 1.79 μM, 11.3 ± 4.13 μM, and 3.36 ± 1.04 μM, respectively.

## 3. Discussion

VRAC is ubiquitously expressed in almost all mammalian cell types and in a wide range of cancer cells [27]. Calcium-activated chloride channel ANO1 is also expressed in various cell types such as smooth muscle, epithelial cells, small sensory neurons, and olfactory-derived cells, and is highly amplified in human cancers such as prostate cancer, breast cancer, esophageal cancer, pancreatic cancer, oral squamous cell carcinoma, and head and neck squamous cell carcinoma [7]. VRAC and ANO1 chloride channels are activated by cell swelling and intracellular calcium increase, respectively. However, previous studies have shown crosstalk between VRAC and ANO1. For example, both LRRC8A and ANO1 are involved in serum-induced VRAC-like currents, and cell swelling and intracellular calcium increase can stimulate both VRAC and ANO1 [28,29]. Therefore, potent and selective inhibitors of VRAC and ANO1 are needed to elucidate the physiological roles of these two chloride channels. For ANO1 inhibitors, as shown in our previous study, Ani9 potently inhibits ANO1 channel activity with only a minimal effect on VRAC [13]. However, all of the previous VRAC inhibitors were nonselective or partial inhibitors [14,15,16,17,18,19,20,21]. Although DCPIB, the most frequently used VRAC inhibitor, did not affect calcium-activated chloride currents in calf pulmonary artery endothelial (CPAE) cells [30], we found that DCPIB potently blocked the ATP-induced intracellular calcium increase associated with the activation of ANO1 (Figure 5E). In addition, DCPIB may also directly affect ANO1 channel activity as it can alter E_act_-activated ANO1 chloride currents (Figure 5F).

DCPIB can activate TREK K^+^ channels in cultured astrocytes and increase basal K^+^ current in neurons in brain slices. In addition, a recent report showed that DCPIB can induce a rapid increase in intracellular Ca^2+^ and directly activate BK channels in a calcium independent manner [23,24]. Here, we showed that DCPIB inhibited hERG K^+^ channel activity at an IC_50_ of 11.4 μM in HEK293T cells expressing hERG. We used FluxOR™ thallium assay which is widely used for the hERG activity test (Figure 6). This result conflicts with a previous report that DCPIB did not inhibit hERG at 10 μM in Xenopus oocytes overexpressed with hERG [25]. This may be due to differences in cellular properties such as the drug permeability of HEK293T cells and oocytes. 

In Figure 7, VI-116 and DCPIB showed inhibition of VRAC activity with different potencies in PC3, HT29, and HeLa cells. These results are consistent with previous reports showing that DCPIB inhibits endogenous VRAC activity with different potency in other cell types [2,31,32]. The difference in potency of VI-116 and DCPIB between cell types is probably due to the different heterogeneous complexes of LRRC8A-E constituting VRAC, and the different drug permeability of each cell type.

Since various cells were cultured in media containing 10% FBS in many in vitro experiments, we measured the inhibitory effect of VI-116 on VRAC activity in the presence of 10% FBS (Figure 3D). In the solution with 10% FBS, the potency of VI-116 was decreased 5.4-fold compared to the condition without FBS. The cause of this phenomenon is presumed to be that VI-116 binds to plasma proteins such as albumin present in FBS, thereby reducing the potency of VI-116.

In summary, the novel VRAC inhibitor VI-116 strongly inhibited VRAC chloride channel activity with only a minimal effect on ANO1, ANO2, and hERG channel activity at 10 μM which completely inhibits VRAC. In addition, VI-116 potently and completely inhibited endogenous VRAC activity in four different cell lines, LN215, PC3, HT29, and HeLa cells. These results suggest that VI-116 can be used as a useful tool for the pharmacological dissection of VRAC, increasing the efficiency of in vitro and in vivo studies to elucidate the physiological role of VRAC.

## 4. Materials and Methods

### 4.1. Cell Culture and Cell Lines

Human ANO1(abc) expressing Fisher rat thyroid (FRT) cells, ANO2 expressing FRT cells, and human CFTR expressing FRT cells were cultured as described in a previous study [13]. LN215 cells were stably transfected with the halide sensor YFP-F46L/H148Q/I152 and NIH-3T3 cells cultured in DMEM were supplemented with 10% FBS, 2 mM L-glutamine, 100 units/mL penicillin, and 100 μg/mL streptomycin. HEK293T cells were stably transfected with the human Kv11.1 (hERG) and cultured in DMEM supplemented with 10% FBS, 2 mM L-glutamine, 100 units/mL penicillin, and 100 μg/mL streptomycin. PC3, HT29, and HeLa cells were stably transfected with the halide sensor YFP-F46L/H148Q/I152 and cultured in RPMI 1640 supplemented with 10% FBS, 2 mM L-glutamine, 100 units/mL penicillin, and 100 μg/mL streptomycin.

### 4.2. Materials and Reagents

DCPIB, T16A_inh_-A01, and other chemicals were purchased from Merck (St. Louis, MO, USA). Chemical libraries used for screening were purchased from ChemDiv (San Diego, CA, USA).

### 4.3. YFP Fluorescence Quenching Assay

YFP-F46L/H148Q/I152L expressed LN215 cells were cultured in 96-well black-walled microplates (Corning Inc., Corning, NY, USA) at a density of 15,000 cells/well in DMEM supplemented with 10% FBS, 100 units/mL penicillin, and 100 μg/mL streptomycin. Screening was performed using FLUOstar Omega microplate reader (BMG Labtech, Ortenberg, Germany). Each well of a 96-well plate was washed 3 times in PBS (200 μL/wash), kept 50 μL hypotonic solution (in mM): 70 NaCl, 5 KCl, 20 HEPES (170 mOsm/kg). Test compounds (0.5 μL) were added to each well at 25 μM final concentration. After 4 min, 96-well plates were transferred to a plate reader for fluorescence assay. Each well was assayed individually for VRAC-mediated I^−^ influx by recording fluorescence continuously (400 ms per point) for 0.4 s (baseline), then 50 μL of 140 mM I^−^ solution was added at 0.5 s, and then YFP fluorescence was recorded for 5 s. The initial iodide influx rate was calculated from fluorescence data by nonlinear regression. 

### 4.4. Apical Membrane Current Measurement

Snapwell inserts containing ANO1(abc) or CFTR-expressing FRT cells were mounted in Ussing chambers (Physiological Instruments, San Diego, CA). The basolateral hemichamber was filled with buffer solution (in mM): 120 NaCl, 5 KCl, 1 MgCl_2_, 1 CaCl_2_, 10 D-glucose, 2.5 HEPES, and 25 NaHCO_3_ (pH 7.4). The apical hemichamber was filled with buffer solution (in mM): 60 NaCl, 60 Na-gluconate, 5 KCl, 1 MgCl_2_, 1 CaCl_2_, 10 D-glucose, 2.5 HEPES, and 25 NaHCO_3_ (pH 7.4). FRT cells were bathed for a 10 min stabilization period and gassed with 5% CO_2_ and 95% O_2_ at 37 °C. The apical membrane current was recorded with an EVC4000 Multi-Channel V/I Clamp (World Precision Instruments, Sarasota, FL, USA) and a PowerLab 4/35 (AD Instruments, Castle Hill, Australia).

### 4.5. FluxOR Potassium Ion Channel Assay

HEK293 cells stably expressing human Kv11.1 (hERG) were cultured in 96-well plates. After 48 h, the cells were incubated at 28 °C for 4 h to increase the membrane expression of hERG. The culture medium was replaced with a 80 μL/well of FluxOR (Invitrogen, Waltham, MA, USA) loading buffer and incubated for 1 h at 37 °C in the dark. After removing the loading buffer, 100 μL of assay buffer was added to each well. To measure the effect of VI-116 and DCPIB on hERG channels, the cells were pretreated with test compounds for 10 min. FluxOR fluorescence (excitation/emission: 490/525 nm) was recorded for 4 s before the addition of 20 μL of stimulus buffer containing thallium ions, and the fluorescence was recorded. FluxOR fluorescence was recorded and analyzed using the FLUOstar Omega microplate reader (BMG Labtech, Ortenberg, Germany) and the MARS Data Analysis Software (BMG Labtech, Ortenberg, Germany). All buffers were prepared according to the manufacturer’s instructions.

### 4.6. Patch-Clamp 

For volume-regulated anion channel (VRAC) currents measurement, the isotonic bath solution contained (in mM): 150 NaCl, 6 KCL, 1.5 CaCl_2_, 1 MgCl_2_, 10 glucose, 1 HEPES, and pH 7.4 with NaOH (320 mOsm). The hypotonic bath solution contained (in mM): 105 NaCl, 6 KCL, 1.5 CaCl_2_, 1 MgCl_2_, 10 glucose, 10 HEPES, and pH 7.4 with NaOH (240 mOsm). The pipette solution contained (in mM): 40 CsCl, 100 Cs-methanesulfonate, 0.5 EGTA, 1 MgCl_2_, 1.9 CaCl_2_, 5 EGTA, 4 Tris−ATP, 10 HEPES, and pH 7.2 with CsOH (290 mOsm). Pipettes were pulled from borosilicate glass and had resistances of 3–5 MΩ after fire polishing. Seal resistances were between 3 and 10 GΩ. The liquid junction potential (~2.4 mV) was not corrected. After establishing the whole-cell configuration, whole-cell capacitance and series resistance were compensated with the amplifier circuitry. VRAC was activated by exposing LN215 cells to a 25% hypotonic bath solution. Whole-cell currents were elicited by applying hyperpolarizing and depolarizing voltage pulses from a holding potential of 0 mV to potentials between −100 mV and +100 mV or −120 mV and +120 mV in steps of 20 mV. Recordings were made at room temperature. Whole-cell recordings were performed using an Axopatch-200B (Axon Instruments, Union City, CA, USA). The current was digitized with a Digidata 1440A converter (Axon Instruments). Data were analyzed using the pCLAMP 10.2 software (Molecular Devices, Sunnyvale, CA, USA).

### 4.7. Cell Proliferation Assay

NIH-3T3 cells were plated on 96-well microplates. After 24 h incubation, cells were treated with VI-116 or DCPIB and incubated for 24 h. The culture medium and the compounds were changed every 12 h. To assess cell proliferation, the cells were incubated with MTS for 1 h. The soluble formazan produced by the cellular reduction of MTS was quantified by measuring the absorbance at 490 nm with an Infinite M200 (Tecan, Grödig, Austria) microplate reader. MTS assay was performed using a CellTiter 96 Aqueous One Solution Cell Proliferation Assay kit (Promega, Madison, WI, USA).

### 4.8. Quantitative PCR Analysis 

The mRNA expression of LRRC8A, B, C, D, and E in HeLa, HT29, LN215, and PC3 cells were measured by qPCR. RNA was isolated using a TRIzol reagent (Invitrogen, Carlsbad, CA, USA), and 500 ng of RNA was used to synthesize complementary DNA (cDNA) using an RNA to cDNA PrimeScript RT-PCR kit (TaKaRa, Shiga, Japan), according to the manufacturer’s protocol. The relative mRNA levels were measured in a StepOnePlus™ Real-Time PCR System (Applied Biosystems, Foster City, CA, USA) using a SYBR Green PCR Master Mix (Applied Biosystems). The primer sequences used were as follows: LRRC8A, sense (5′-GGGTTGAACCATGATTCCGGTGAC-3′) and antisense (5′-GAAGACGGCAATCATCAGCATGAC-3′); LRRC8B, sense (5′-CCTGGATGGCCCACAGGTAATAG-3′) and antisense (5′-ATGCTGGTCAACTGGAACCTCTGC-3′); LRRC8C, sense (5′-ACAAGCCATGAGCAGCGAC-3′) and antisense (5′-GGAATCATGTTTCTCCGGGC-3′); LRRC8D, sense (5′-ATGGAGGAGTGAAGTCTCCTGTCG-3′) and antisense (5′-CTTCCGCAAGGGTAAACATTCCTG-3′); and LRRC8E, sense (5′-ACCGTGGCCATGCTCATGATTG-3′) and antisense (5′-ATCTTGTCCTGTGTCACCTGGAG-3′). The mRNA levels of LRR8A-E were normalized to β-actin.

### 4.9. Intracellular Calcium Measurement 

FRT cells were plated on 96-well black-walled microplates and loaded with Fluo-4 NW according to the manufacturer’s protocol (Invitrogen, Carlsbad, CA, USA). After 1 h incubation, the cells were added with VI-116 or DCPIB and incubated for 10 min. The 96-well plates were transferred to a plate reader for fluorescence assay. Fluo-4 fluorescence (excitation/emission: 485/538 nm) was measured with a FLUOstar Omega microplate reader (BMG Labtech). Intracellular calcium was increased by an application of 100 μM ATP.

### 4.10. Statistical Analysis

The results of the experiments are displayed as the means ± S.E. The statistical analysis was performed with a Student’s *t*-test. A value of *p* < 0.01 was considered statistically significant.

## Figures and Tables

**Figure 1 ijms-23-05168-f001:**
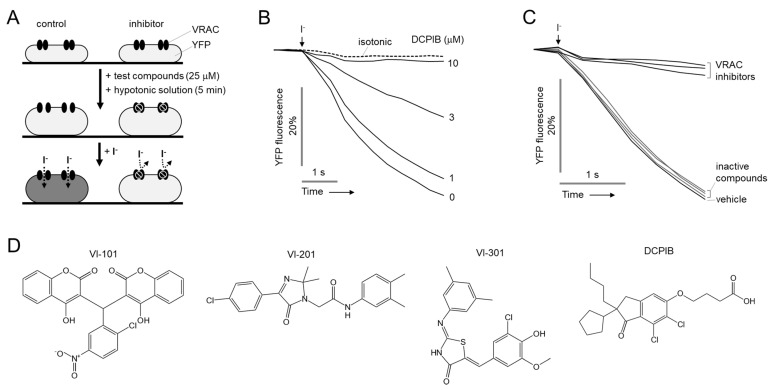
Identification of novel VRAC inhibitors. (**A**) Principle of high−throughput screening assay. (**B**) Representative YFP fluorescence traces for VRAC activity. LN215 cells were treated with indicated concentrations of DCPIB for 5 min. VRAC was activated by application of hypertonic solution for 5 min. (**C**) Representative YFP fluorescence traces for VRAC inhibitors and inactive compounds. (**D**) Chemical structures of novel VRAC inhibitors (VI-101, VI-201, and VI-301) and DCPIB.

**Figure 2 ijms-23-05168-f002:**
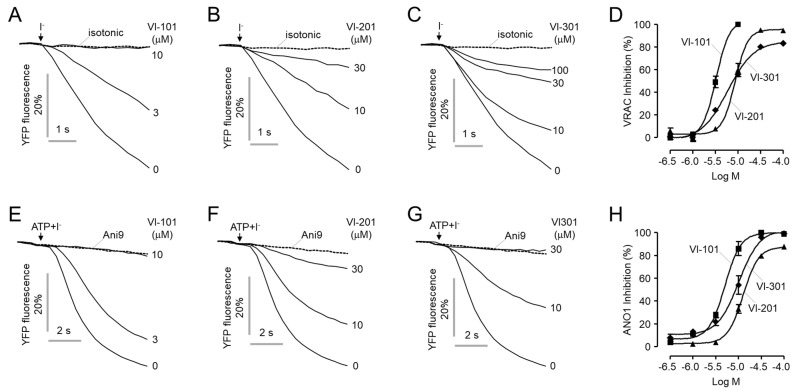
Inhibitory effects of VI-101, VI-201 and VI-301 on VRAC and ANO1. (**A**–**C**) Representative YFP fluorescence traces for VRAC activity. LN215 cells were treated with the indicated concentrations of VI-101, VI-201 and VI-301 with hypotonic solution for 5 min. (**D**) Dose-response curve of VRAC inhibition (mean ± S.E., *n* = 4). (**E**–**G**) Representative YFP fluorescence traces for ANO1 activity. FRT−ANO1−YFP cells were treated with the indicated concentrations of VI-101, VI-201 and VI-301 for 10 min. ANO1 was activated and inhibited by ATP (100 μM) and Ani9 (10 μM, dashed line), respectively. (**H**) Dose-response curve of ANO1 inhibition (mean ± S.E., *n* = 4).

**Figure 3 ijms-23-05168-f003:**
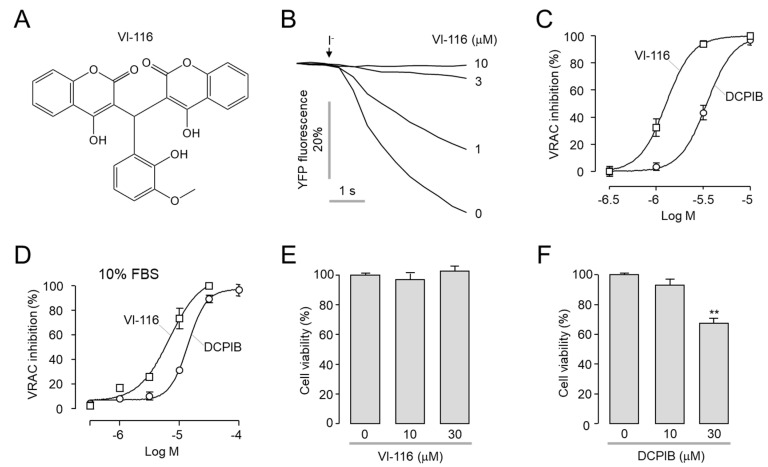
Effect of VI-116 and DCPIB on VRAC activity and cell viability. (**A**) Structure of VI-116. (**B**) Representative YFP fluorescence traces. The inhibitory effects of VI-116 on VRAC activity were determined using YFP fluorescence assay in LN215 cells. VRAC was activated by application of hypertonic solution for 5 min. (**C**) Summary of VRAC dose-response (mean ± S.E., *n* = 4). (**D**) Summary of VRAC dose−response with 10% FBS (mean ± S.E., *n* = 4). (**E**,**F**) NIH3T3 cells were treated with the indicated concentrations of VI-116 or DCPIB for 24 h and cell viability was measured by MTT assay (mean ± S.E., *n* = 4), ** *p* < 0.01.

**Figure 4 ijms-23-05168-f004:**
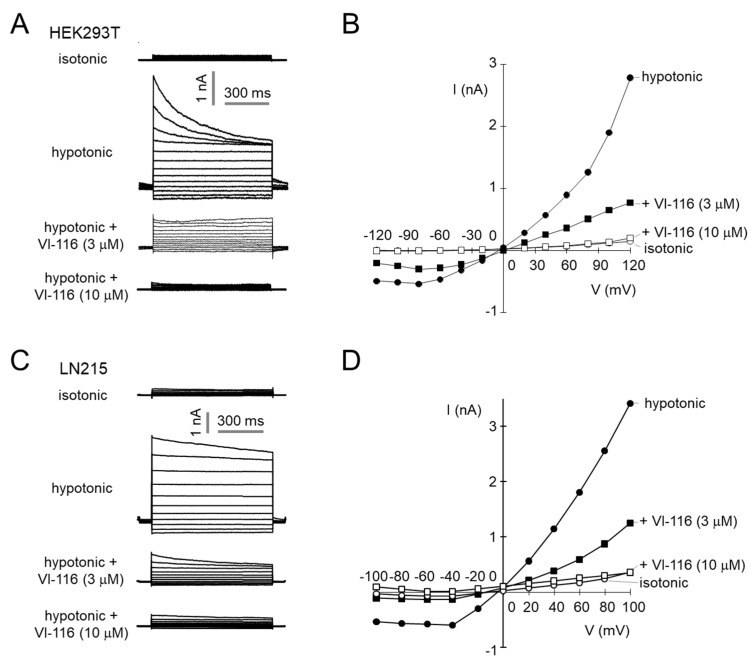
VI-116 potently blocks VRAC−mediated chloride current in HEK293T and LN215 cells. (**A**) Whole-cell membrane currents were recorded at a holding potential of −50 mV by voltage steps ranging from −120 to 120 mV (in steps of 20 mV) in HEK293T cells. VRAC was activated by application of hypertonic solution for 3 min, and then the cells were treated with indicated concentrations of VI-116. (**B**) Current/voltage plot of mean currents at the middle of each voltage pulse (*n* = 3). (**C**) Whole-cell VRAC currents were recorded at a holding potential of −50 mV by voltage steps ranging from −100 to 100 mV (in steps of 20 mV) in LN215 cells. (**D**) Current/voltage plot of mean currents at the middle of each voltage pulse (*n* = 3).

**Figure 5 ijms-23-05168-f005:**
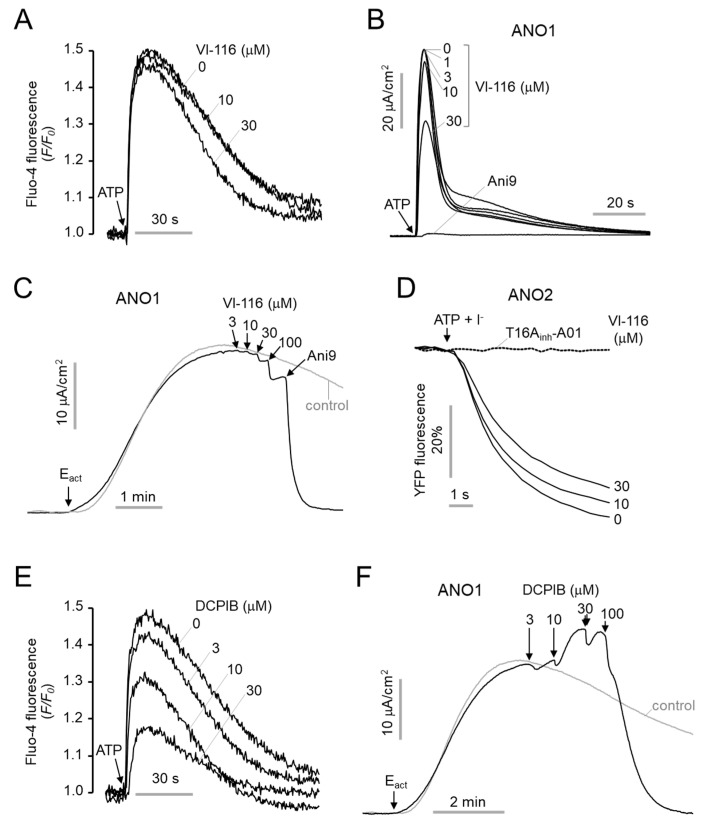
Effects of VI-116 on ANO1 and ANO2 chloride channel activity. (**A**) Representative Fluo4 fluorescence traces from 3 independent experiments in FRT cells. Cells were pretreated with VI-116 for 10 min, and P2Y receptors were activated with 100 μM ATP. (**B**,**C**) Representative apical membrane currents from 3–4 independent experiments in FRT cells expressing human ANO1. (**B**) Cells were treated with indicated concentrations of VI-116 and DCPIB for 10 min, and ANO1 was fully activated by 100 μM ATP and completely blocked by 10 μM Ani9. (**C**) ANO1 was activated with 3 μM E_act_ and the indicated concentrations of VI-116 were applied, and remaining ANO1 channel activity was completely blocked by 10 μM Ani9. (**D**) Representative YFP fluorescence traces in FRT cells expressing human ANO2. Cells were treated with the indicated concentrations of VI-116 for 10 min. ANO2 was fully activated by 100 μM ATP and completely blocked by 30 μM T16A_inh_-A01. (**E**) Representative Fluo4 fluorescence traces from 3 independent experiments in FRT cells. Cells were pretreated with DCPIB for 10 min followed by application of 100 μM ATP. (**F**) Representative apical membrane currents in FRT cells expressing human ANO1. ANO1 was activated with 3 μM E_act_ and then the indicated concentrations of DCPIB were applied.

**Figure 6 ijms-23-05168-f006:**
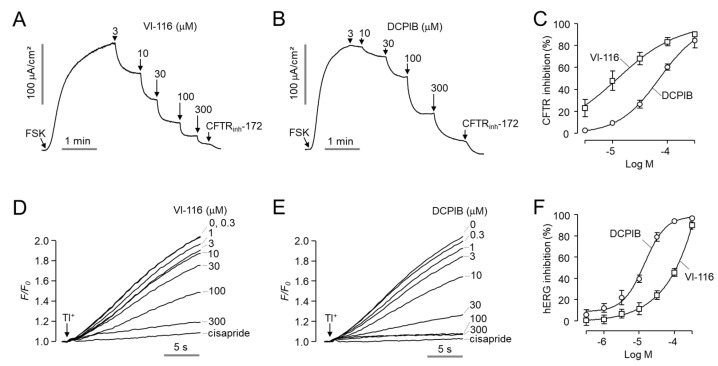
Effects of VI-116 and DCPIB on CFTR and hERG channel activity. (**A**,**B**) Representative apical membrane currents from 5 independent experiments in FRT cells expressing human CFTR. CFTR was fully activated by 20 μM forskolin (FSK) and inhibited with indicated concentrations of VI-116 and DCPIB. CFTR was completely blocked by 20 μM CFTR_inh_−172. (**C**) Summary of dose-response (mean ± S.E., *n* = 5). (**D**,**E**) The effect of VI-116 and DCPIB on hERG activity was measured using a thallium flux assay in HEK293T cells expressing hERG. Cells were treated with the indicated concentrations of VI-116 and DCPIB for 10 min. hERG channel was activated by application of stimulus buffer and was inhibited by 50 μM cisapride. (**F**) Summary of dose−response (mean ± S.E., *n* = 6).

**Figure 7 ijms-23-05168-f007:**
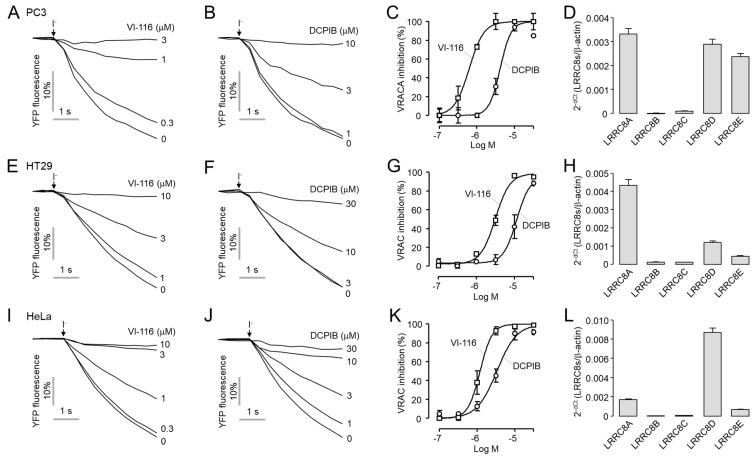
Effects of VI-116 and DCPIB on VRAC activity in PC3, HT29 and HeLa cells. (**A**,**B**) Representative YFP fluorescence traces for the inhibitory effects of VI-116 and DCPIB in PC3 cells. VRAC was activated by application of hypertonic solution for 5 min. (**C**) Summary of dose-response (mean ± S.E., *n* = 4). (**D**) LRRC8A−E mRNA expression levels in PC3 cells. (**E**,**F**) Inhibitory effects of VI-116 and DCPIB in HT29 cells. VRAC was activated by application of hypertonic solution for 5 min. (**G**) Summary of dose-response (mean ± S.E., *n* = 4). (**H**) LRRC8A−E mRNA expression levels in HT29 cells. (**I**,**J**) Inhibitory effects of VI-116 and DCPIB in HeLa cells. VRAC was activated by application of hypertonic solution for 5 min. (**K**) Summary of dose-response (mean ± S.E., *n* = 4). (**L**) LRRC8A−E mRNA expression levels in HeLa cells. mRNA expression levels of LRRC8A−E were determined by qRT−PCR and normalized to β−actin (mean ± S.E., *n* = 4).

**Table 1 ijms-23-05168-t001:** Effect of VI-101 derivatives on VRAC and ANO1 activity. IC_50_ values were determined using YFP fluorescence quenching assay in LN215 and FRT-ANO1 cells (*n* = 4).

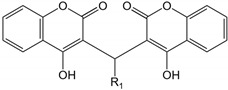
ID	R_1_	VRACIC_50_ (μM)	ANO1IC_50_ (μM)	ID	R_1_	VRACIC_50_ (μM)	ANO1IC_50_ (μM)
VI-101	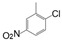	3.3	3.2	VI-111	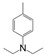	2.9	3.2
VI-102	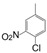	0.8	3.0	VI-112	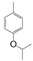	1.4	0.8
VI-103	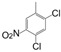	3.3	3.2	VI-113	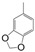	1.9	3.9
VI-104	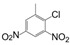	30.0	37.9	VI-114	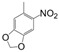	10.6	20.3
VI-105	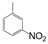	4.9	3.8	VI-115	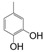	47.9	61.6
VI-106	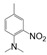	1.3	3.6	VI-116	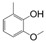	1.3	39.1
VI-107	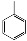	103.3	95.2	VI-117	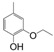	6.8	14.2
VI-108	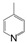	17.0	34.7	VI-118	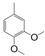	4.0	14.2
VI-109	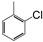	0.9	2.0	VI-119	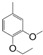	12.0	4.9
VI-110	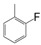	0.6	4.8	VI-120	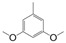	9.3	3.9

## Data Availability

Not applicable.

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
