# Peer review of "VI-116, A Novel Potent Inhibitor of VRAC with Minimal Effect on ANO1"

_ijms, 2022, doi:10.3390/ijms23095168_

Round 1

Reviewer 1 Report

In this pharmacological study, using high-throughput screening the authors have identify a candidate molecule to target the ubiquitously express VRAC/LRRC8 channel (volume regulated anion channel). Historically, this conductance was shown to be involve in regulatory volume decrease but a growing number of evidence suggest its contribution to various physiological process. They test using various approaches (flux of Iodide trough chloride channel and quenching of a fluorescent biomarker, patch-clamp and transepithelial current measurements) the efficacy of some derivative of the initial candidate molecule on VRAC conductance (LRRC8), CaCC (ANO-1 or 2) and hERG K currents. They finally conclude that the VI-116 is the most potent to inhibits VRAC with only moderate effect on the other tested currents.

Altogether, this interesting study might help to identify a useful tool for the pharmacological  dissection of VRAC function (this is essential considering the moderate specificity of the commercial chloride channels inhibitors).

The manuscript is clearly written but a bit superficial concerning the description and the contribution of the chloride channels explored in this study. Majors’ experiments are also needed to confirm and explore with more details the efficacy and the selectivity of the new drug identified: sensitivity to CFTR, a better characterisation of the effect on ANO-1/2 using patch-clamp,….altogether, this study is too preliminary to be publish in its current form.

Majors:

1/ Introduction concerning the dual expression of VRAC and ANO1 1 channels is not enough precise. While VRAC and LRRC8 family are ubiquitously express ANO1 is restricted at least in human to some specific cells generally epithelial cells. Ano 2 is even lower expressed and limited to a restricted number of tissues. Therefore, extrapolating the putative combination of both channels for cell volume regulation is a bit confusing despite some data from the literature. Please enhanced the difference between both type of channels: they are activated by very different stimulus and probably contribute to distinct functions. While ANO1 is clearly sensitive to intracellular calcium increase, the calcium sensitivity of VRAC is not fully elucidated and probably indirect (for example, increasing intracellular calcium by ionomycin induced a very weak activation of VRAC) this is clearly not the main trigger. This has to be stated somewhere.

2/ I fully disagree with this sentence: …”(DCPIB), tamoxifen, mibefradil, mefloquine, clomiphene, nafoxidine and carbenoxolone, pranlukast and zafirlukast was identified but none of these VRAC inhibitors are potent or selective”. Most of these inhibitors are probably aspecific but most of then and especially DCPIB are inhibiting the VRAC current. They are “potent” despite the high concentrations needed for some of them. For example, IC50 for DCPIB was founded around 3-5 µM in the same µMolar range of VI-116.

3/ A major criticism is the cell model used: FRT-ANO1 cells are pancreatic cells that are classically used to study CFTR chloride channels activity. Therefore, it is probably not the optimal cells because this cell type expresses simultaneously at least 3 different chloride channels: VRAC/LRRC8, CFTR/ABCC7 and CaCC/ANO1. It might be very difficult to discriminate between the different type of channels in the different experimental conditions used.

4/ Based on the previous comment a second major criticism is the absence of any test of the VI-116 compounds on CFTR channels. To use this new compound as a gold standard for VRAC inhibition, the authors must test it on CFTR currents. This is essential.

5/ Test of Vi-116 on ANO-1 transepithelial current: Most of the experiments using the new drug were performed by measurement trans epithelial current following ATP exposure (this is an indirect measure). The way of activation by ATP is based on a complex signalling involving activation of purinoceptors that in turn induce and influx of calcium that finally activate CaCC. We cannot exclude that this new drug can act on purinergic receptors. The best way to exclude side effect will be to test the drug on CaCC/Ano1 conductance in whole-cell configuration following intracellular increase of Calcium (ionomycin or A23187 treatment). A second major point for this part is linked to the ubiquitous expression of LRRC8/VRAC: how to exclude a cross talk between LRRC8/VRAC and CaCC/ANO1? Inhibition of LRRC8 by VI-116 would then affect ANO1 current ? Experiments must be done in cell that do no express LRRC8A subunit for example but ANO-1 to be sure that the current measured is “purely” mediated by ANO-1 channels activation.

6/ The inhibition of CaCC by DCPIB given by the authors is inconsistent with the literature (see decher 2001), explanation should be given to explain these surprising results.

7/ VI-116 potently inhibits endogenous VRAC activity in PC3, HT29 and HeLa cells:The authors speculate that the difference of sensitivity to VI-116 or DCPIB is related to a specific combination of the different LRRC8 subunits. It is clearly attractive and plausible but highly speculative, the authors give no data about the real level of expression of each subunit in the different cell line used.

Minor points

What is the rational of testing VI-116 on human ether-a-go-go (hERG) K channel? Please explain.

Patch-clamp: The rational of using 40 CsCl, 100 Cs-methanesulfonate is not given. please explain why the pipette solution contains methanesulfonate. If we consider only the chloride ions concentration of the bath and pipette solutions, the reversal potential on the I/V curve should not be at 0 mv but at -24 mV. please explain this discrepancy.

Author Response

We greatly appreciate the editor’s and reviewers’ efforts to carefully review our manuscript and the valuable comments and suggestions offered for the improvement of the manuscript (ijms-1661458). We have made each of the suggested revisions. The points of criticism raised by the reviewers were addressed by a point-by-point response. Changes in the manuscript text are highlighted in red color font.

Reviewer #1

In this pharmacological study, using high-throughput screening the authors have identify a candidate molecule to target the ubiquitously express VRAC/LRRC8 channel (volume regulated anion channel). Historically, this conductance was shown to be involve in regulatory volume decrease but a growing number of evidence suggest its contribution to various physiological process. They test using various approaches (flux of Iodide trough chloride channel and quenching of a fluorescent biomarker, patch-clamp and transepithelial current measurements) the efficacy of some derivative of the initial candidate molecule on VRAC conductance (LRRC8), CaCC (ANO-1 or 2) and hERG K currents. They finally conclude that the VI-116 is the most potent to inhibits VRAC with only moderate effect on the other tested currents.

Altogether, this interesting study might help to identify a useful tool for the pharmacological dissection of VRAC function (this is essential considering the moderate specificity of the commercial chloride channels inhibitors).

The manuscript is clearly written but a bit superficial concerning the description and the contribution of the chloride channels explored in this study. Majors’ experiments are also needed to confirm and explore with more details the efficacy and the selectivity of the new drug identified: sensitivity to CFTR, a better characterization of the effect on ANO-1/2 using patch-clamp,….altogether, this study is too preliminary to be publish in its current form..

Major Comments:

  1. Introduction concerning the dual expression of VRAC and ANO1 1 channels is not enough precise. While VRAC and LRRC8 family are ubiquitously express ANO1 is restricted at least in human to some specific cells generally epithelial cells. ANO 2 is even lower expressed and limited to a restricted number of tissues. Therefore, extrapolating the putative combination of both channels for cell volume regulation is a bit confusing despite some data from the literature. Please enhanced the difference between both type of channels: they are activated by very different stimulus and probably contribute to distinct functions. While ANO1 is clearly sensitive to intracellular calcium increase, the calcium sensitivity of VRAC is not fully elucidated and probably indirect (for example, increasing intracellular calcium by ionomycin induced a very weak activation of VRAC) this is clearly not the main trigger. This has to be stated somewhere.

Response: Thank you for the valuable comments. We have more accurately described the expression patterns and physiological characteristics of VRAC and ANO1 in the revised introduction.

  1. I fully disagree with this sentence: …”(DCPIB), tamoxifen, mibefradil, mefloquine, clomiphene, nafoxidine and carbenoxolone, pranlukast and zafirlukast was identified but none of these VRAC inhibitors are potent or selective”. Most of these inhibitors are probably aspecific but most of them and especially DCPIB are inhibiting the VRAC current. They are “potent” despite the high concentrations needed for some of them. For example, IC50 for DCPIB was founded around 3-5 µM in the same µMolar range of VI-116.

Response: Thank you for the valuable comments. We agree with the reviewer's comments and have revised the sentence to be more precise.

  1. A major criticism is the cell model used: FRT-ANO1 cells are pancreatic cells that are classically used to study CFTR chloride channels activity. Therefore, it is probably not the optimal cells because this cell type expresses simultaneously at least 3 different chloride channels: VRAC/LRRC8, CFTR/ABCC7 and CaCC/ANO1. It might be very difficult to discriminate between the different type of channels in the different experimental conditions used.

Response: FRT cells are poorly differentiated epithelial cell line derived from Fischer rat thyroid gland, which endogenously express VRAC but not CFTR and ANO1. The FRT-ANO1 cells were established by stably transfection of human ANO1 (abc isoform) and do not express CFTR. Because FRT cells grow well and form tight junctions, they are very useful for short-circuit current measurement. In this study, we used FRT-ANO1 cells to test the effect of VRAC inhibitors on ANO1 channel activity.

  1. Based on the previous comment a second major criticism is the absence of any test of the VI-116 compounds on CFTR channels. To use this new compound as a gold standard for VRAC inhibition, the authors must test it on CFTR currents. This is essential.

Response: Thank you for the valuable comments. We have investigated the effect of VI-116 and DCPIB on CFTR channel activity in FRT-CFTR cells stably expressing human wild-type CFTR. As shown in Figure 6, VI-116 and DCPIB inhibited CFTR chloride channel activity with IC50 values of 12.4 mM and 71.7 mM, respectively.

  1. Test of VI-116 on ANO-1 transepithelial current: Most of the experiments using the new drug were performed by measurement trans epithelial current following ATP exposure (this is an indirect measure). The way of activation by ATP is based on a complex signaling involving activation of purinoceptors that in turn induce and influx of calcium that finally activate CaCC. We cannot exclude that this new drug can act on purinergic receptors. The best way to exclude side effect will be to test the drug on CaCC/Ano1 conductance in whole-cell configuration following intracellular increase of Calcium (ionomycin or A23187 treatment). A second major point for this part is linked to the ubiquitous expression of LRRC8/VRAC: how to exclude a cross talk between LRRC8/VRAC and CaCC/ANO1? Inhibition of LRRC8 by VI-116 would then affect ANO1 current? Experiments must be done in cell that do no express LRRC8A subunit for example but ANO-1 to be sure that the current measured is “purely” mediated by ANO-1 channels activation.

Response: In FRT-ANO1 cells, ATP-induced chloride channel activity is solely due to ANO1 and not VRAC. This is proven by the complete inhibition of ATP-induced chloride channel activity by Ani9, an ANO1-specific inhibitor that does not inhibit VRAC. In addition, several previous studies show that ATP-induced chloride channel activity in FRT-ANO1 cells is solely due to ANO1 (J Biol Chem. 2011 Jan 21;286(3):2365-74; PLoS One. 2016 May 24;11(5):e0155771).

  1. The inhibition of CaCC by DCPIB given by the authors is inconsistent with the literature (see decher 2001), explanation should be given to explain these surprising results.

Response: Thank you for the valuable comments. We were also surprised to find that DCPIB inhibits ANO1 well, and this result was one of the reasons for developing a VRAC inhibitor that does not inhibit ANO1. In original study of DCPIB, the authors observed effect of DCPIB on bovine CaCC activity not human CaCC/ANO1. In addition, we found that DCPIB potently inhibited the ATP-induced increase in intracellular calcium concentration. This is the main reason why DCPIB strongly blocked ATP-induced ANO1 activation. And for this reason, an additional experiment was conducted to find out whether DCPIB directly activates ANO1 using Eact, an ANO1-specific activator. As shown in Figure 5E, DCPIB strongly blocked ATP-induced increase in intracellular calcium levels in a dose-dependent manner. In addition, DCPIB had a unique effect on Eact-activated ANO1 chloride channel activity. At 10 mM, ANO1 chloride current activated by Eact was increased, but at 100 mM, ANO1 chloride current was almost completely inhibited (Figure 5F).

  1. VI-116 potently inhibits endogenous VRAC activity in PC3, HT29 and HeLa cells: The authors speculate that the difference of sensitivity to VI-116 or DCPIB is related to a specific combination of the different LRRC8 subunits. It is clearly attractive and plausible but highly speculative, the authors give no data about the real level of expression of each subunit in the different cell line used.

Response: Thank you for the valuable comments. We have performed real-time PCR for the quantification of mRNA expression levels of LRRC8 subunits in PC3, HT29 and HeLa cells. As shown in Figure 7, qRT-PCR analysis revealed that PC3 cells had relatively high expression rates of LRRC8A, D, and E, but HT26 cells and HeLa cells had relatively high expression rates of LRRC8A and LRRC8D, respectively.

Minor Comments:

  1. What is the rational of testing VI-116 on human ether-a-go-go (hERG) K+ channel? Please explain.

Response:  In this study, we observed the effect of VI-116 on hERG K+ channel activity because the hERG K+ channel is a major anti-target of drug discovery. Blockage of hERG channels causes long QT syndrome and increases the risk of cardiac arrhythmias and sudden death. In addition, there were some reports that DCPIB had an effect on the activity of some K+ channels, so we measured the effect of VI-116 on hERG K+ channel activity using a method for evaluating hERG K+ activity previously set up in our laboratory. The rational of testing the effect of VRAC inhibitors on hERG channel activity was corrected in the revised manuscript.

  1. Patch-clamp: The rational of using 40 mM CsCl, 100 mM Cs-methanesulfonate is not given. please explain why the pipette solution contains methanesulfonate. If we consider only the chloride ions concentration of the bath and pipette solutions, the reversal potential on the I/V curve should not be at 0 mv but at -24 mV. please explain this discrepancy.

Response: The VRAC patch clamp was performed with reference to the previous paper, and a pipette solution containing Cs-methanesulfonate was used in the previous paper (Science. 2014 May 9;344(6184):634-8. doi: 10.1126/science.1252826.).

Reviewer 2 Report

Using a high-throughput screening the  authors discovered a novel VRAC inhibitor, which is about 13-fold less potent on the TMEM16A calcium activated chloride channel (I suggest to use the TMEM16 nomenclature because “ANO” is a misnomer in that the protein does not have 8 TMDs).

While this result is laudable several important control measurements are missing as outlined in the specific comments below.

  1. DCPIB effects are strongly reduced in the presence of serum. Since possible applications of VI-116 are supposed to be conducted in the presence of serum, it’s effect on efficacy needs to be determined.

  1. Effects of VI-116 on TMEM16A have only been studied using indirect assays. Dose-response using patch-clamp has to be performed. This may reveal also voltage-dependence of block.

  1. Line 38: In the introduction, the heteromeric nature of VRAC has to be immediately introduced.

  1. Line 40. Results reported in references 6 and 7 are not credible and have not been reproduced by other laboratories.

  1. Line 54: “Ani9, a potent …”

  1. Lines 62: The activation of BK by DCPIB should be introduced in the introduction.

  1. Lines 63: In addition to activating BK, DCPIB has been shown to increase calcium. This needs to be mentioned in the introduction.

  1. Line 70: change “with halide 70 sensors YFP-F46L/H148Q/I152L” in “with halide 70 sensor YFP-F46L/H148Q/I152L” and provide ref. to this sensor.

  1. Fig. 1B, C: what is the y-axis showing exactly?

  1. Table 1: how have the EC50s been determined?

  1. Line 128: “Minimal “

  1. Effects on TREK and BK channels have not been tested. Either the compound has to be made freely available such that other groups could test that, or the authors need to do these experiments.

  1. Which chemical libraries have been used exactly?

Author Response

We greatly appreciate the editor’s and reviewers’ efforts to carefully review our manuscript and the valuable comments and suggestions offered for the improvement of the manuscript (ijms-1661458). We have made each of the suggested revisions. The points of criticism raised by the reviewers were addressed by a point-by-point response. Changes in the manuscript text are highlighted in red color font.

Reviewer #2

Using a high-throughput screening the authors discovered a novel VRAC inhibitor, which is about 13-fold less potent on the TMEM16A calcium activated chloride channel (I suggest to use the TMEM16 nomenclature because “ANO” is a misnomer in that the protein does not have 8 TMDs). While this result is laudable several important control measurements are missing as outlined in the specific comments below.

Comments:

  1. DCPIB effects are strongly reduced in the presence of serum. Since possible applications of VI-116 are supposed to be conducted in the presence of serum, it’s effect on efficacy needs to be determined.

Response: Thank you for the valuable comments. Additional experiment was performed to investigate the effect of serum on the potency of VI-116 and DCPIB. As shown in Figure 3D, in the presence of 10% FBS, VI-116 and DCPIB inhibited VRAC chloride channel activity with IC50 6.89 μM and 14.1 μM, respectively

  1. Effects of VI-116 on TMEM16A have only been studied using indirect assays. Dose-response using patch-clamp has to be performed. This may reveal also voltage-dependence of block.

Response: In Figure 5B-C, we showed potent inhibition of apical membrane current of ANO1 by high concentration of VI-116 in FRT-ANO1 cells. In this study, we did not perform a patch clamp study to show the voltage-dependent blockade of ANO1 by VI-116 because VI-116 did not inhibit ANO1 at concentrations (3~10 mM) that fully inhibited VRAC.

  1. Line 38: In the introduction, the heteromeric nature of VRAC has to be immediately introduced.

Response: Thank you. The heterogeneous nature of VRAC is described at the beginning of the introduction.

  1. Line 40. Results reported in references 6 and 7 are not credible and have not been reproduced by other laboratories.

Response: References 6 and 7 have been deleted and the text has been corrected.

  1. Line 54: “Ani9, a potent …”

Response: Thank you. Corrected.

  1. Lines 62: The activation of BK by DCPIB should be introduced in the introduction.

Response: Thank you. The activation of BK by DCPIB has been described in the introduction.

  1. Lines 63: In addition to activating BK, DCPIB has been shown to increase calcium. This needs to be mentioned in the introduction.

Response: Thank you. It has been described in the introduction.

  1. Line 70: change “with halide 70 sensors YFP-F46L/H148Q/I152L” in “with halide 70 sensor YFP-F46L/H148Q/I152L” and provide ref. to this sensor.

Response: Thank you. Corrected.

  1. Fig. 1B, C: what is the y-axis showing exactly?

Response: Yes, the rates of VRAC-induced YFP fluorescence quenching vary between cell batches.

  1. Table 1: how have the IC50s been determined?

Response: The IC50 values were determined using YFP fluorescence quenching assay in LN215 and FRT-ANO1 cells (n = 4).

  1. Line 128: “Minimal “

Response: Thank you. Corrected.

  1. Effects on TREK and BK channels have not been tested. Either the compound has to be made freely available such that other groups could test that, or the authors need to do these experiments.

Response: The compound is freely available for other groups.

  1. Which chemical libraries have been used exactly?

Response: We used a commercial library containing 55,000 drug-like synthetic compounds from ChemDiv Inc.

Reviewer 3 Report

The authors of the manuscript present a novel inhibitor (VI-116) of volume-regulated anion channel (VRAC) that excert minimal effect on calcium activated chloride channel ANO1. VRAC channels are the key players in vertebrates cell volume regulation that transport chloride ions and various organic osmolytes. Most of known VRAC inhibitors are non  potent or selective (e.g. DCPIB targets also different ion transporting proteins such as H/K-ATPase, Kir channels, etc. and supress mitochondrial respiration). Thus, there is a need to use  selective inhibitors of VRAC to study its physiological function. 

Overall I find the manuscript very interesting and easy to follow. I have only few comments to the authors:

1) I do not see the information about cytotoxicity of VI-116. 

2) Subsection "2.4. Minial effect of VI-116 on human ANO1, ANO2 and hERG channel activity"

I do not fully understand why the authors decided to study the influence of inhibitor on hERG channel activity? 

In my opinion it would be better to check how VI-116 affects the activity of different potassium channels. Since the author focused on ANO1 and ANO2 (calcium-activated chloride channels) it would be better to focus on calcium activated K+ channels. 

It would be also good to check how VI-116 influences the intracellular Ca2+

The information about the effects of VI-116 on CFTR channel activity and mitochondrial respiration would be an asset. 

3) Subsection "4.3. YFP fluorescence quenching assay"

line 239:  "hypotonic solution (in mM): 70 NaCl, 5 KCl, 20 HEPES, 120 mannitol (150 mOsm/kg)" - please check the composition of solution, for me it is not hypotonic

Author Response

We greatly appreciate the editor’s and reviewers’ efforts to carefully review our manuscript and the valuable comments and suggestions offered for the improvement of the manuscript (ijms-1661458). We have made each of the suggested revisions. The points of criticism raised by the reviewers were addressed by a point-by-point response. Changes in the manuscript text are highlighted in red color font.

Reviewer #3:

The authors of the manuscript present a novel inhibitor (VI-116) of volume-regulated anion channel (VRAC) that excert minimal effect on calcium activated chloride channel ANO1. VRAC channels are the key players in vertebrates cell volume regulation that transport chloride ions and various organic osmolytes. Most of known VRAC inhibitors are non potent or selective (e.g. DCPIB targets also different ion transporting proteins such as H/K-ATPase, Kir channels, etc. and supress mitochondrial respiration). Thus, there is a need to use selective inhibitors of VRAC to study its physiological function. Overall, I find the manuscript very interesting and easy to follow. I have only few comments to the authors:

Comments:

  1. I do not see the information about cytotoxicity of VI-116.

Response: Thank you for the valuable comments. Additional experiment was performed to investigate the effect of VI-116 on intracellular calcium levels. To determine whether VI-116 and DCPIB are cytotoxic, we observed the effect of VI-116 and DCPIB on cell viability in NIH-3T3 cells. As shown in Figure 3E-F, VI-116 did not affect cell viability up to 30 mM, but DCPIB significantly reduced cell viability at 30 mM.

  1. Subsection "2.4. Minimal effect of VI-116 on human ANO1, ANO2 and hERG channel activity" I do not fully understand why the authors decided to study the influence of inhibitor on hERG channel activity? In my opinion it would be better to check how VI-116 affects the activity of different potassium channels. Since the author focused on ANO1 and ANO2 (calcium-activated chloride channels) it would be better to focus on calcium activated K+ channels.

Response:  In this study, we observed the effect of VI-116 on hERG K+ channel activity because the hERG K+ channel is a major anti-target of drug discovery. Blockage of hERG channels causes long QT syndrome and increases the risk of cardiac arrhythmias and sudden death. In addition, there were some reports that DCPIB had an effect on the activity of some K+ channels, so we measured the effect of VI-116 on hERG K+ channel activity using a method for evaluating hERG K+ activity previously set up in our laboratory.

  1. It would be also good to check how VI-116 influences the intracellular Ca2+

Response: Thank you for the valuable comments. Additional experiment was performed to investigate the effect of VI-116 on intracellular calcium levels. we measure the effect of VI-116 on intracellular calcium signaling in FTR cells. As shown in Figure 5A, VI-116 did not affect the ATP-induced increase in intracellular calcium levels up to 10 mM and showed a weak inhibitory effect on intracellular calcium signaling at 30 mM.

  1. The information about the effects of VI-116 on CFTR channel activity and mitochondrial respiration would be an asset.

Response: Thank you for the valuable comments. We have investigated the effect of VI-116 and DCPIB on CFTR channel activity in FRT-CFTR cells stably expressing human wild-type CFTR. As shown in Figure 7, VI-116 and DCPIB inhibited CFTR chloride channel activity with IC50 12.4 μM and 71.7 μM, respectively. Unfortunately, we cannot measure the effect of VI-116 on mitochondrial respiration.

  1. Subsection "4.3. YFP fluorescence quenching assay"; line 239: "hypotonic solution (in mM): 70 NaCl, 5 KCl, 20 HEPES, 120 mannitol (150 mOsm/kg)" - please check the composition of solution, for me it is not hypotonic

Response:  Thank you. Corrected.

Round 2

Reviewer 2 Report

Most issues have been adequately addressed. The following three points require however some revision.

1. The discussion has to comment on the drastic effects of serum on the inhibition, and to underline the consequent limitations of the compound. The impact in various scenarios of application need to be discussed.

2. The ANO nomenclature has to be changed to the TMEM16 nomenclature.

3. Regarding my original comment

“Lines 63: In addition to activating BK, DCPIB has been shown to increase calcium. This needs to be mentioned in the introduction.”

the authors responded:

“Response: Thank you. It has been described in the introduction.”

However, I could not find this description in the revised version.

Author Response

We greatly appreciate the editor’s and reviewers’ efforts to carefully review our manuscript and the valuable comments and suggestions offered for the improvement of the manuscript (ijms-1672626). We have made each of the suggested revisions. The points of criticism raised by the reviewers were addressed by a point-by-point response. Changes in the manuscript text are highlighted in red color font.

Reviewer #2:

Most issues have been adequately addressed. The following three points require however some revision.

  1. The discussion has to comment on the drastic effects of serum on the inhibition, and to underline the consequent limitations of the compound. The impact in various scenarios of application need to be discussed.

Response: Thank you for your valuable comments. The drastic effect of serum on VI-116-mediated VRAC inhibition was discussed in the revised discussion section.

  1. The ANO nomenclature has to be changed to the TMEM16 nomenclature.

Response: Thank you. Since both ANO and TMEM16 nomenclature are commonly used, we added this information to the introduction.

  1. Regarding my original comment

“Lines 63: In addition to activating BK, DCPIB has been shown to increase calcium. This needs to be mentioned in the introduction.” the authors responded: “Response: Thank you. It has been described in the introduction.” However, I could not find this description in the revised version.

Response: Sorry for the mistake. The effect of DCPIB on intracellular calcium concentration is described in the revised introduction.